# An Integrated Data-Driven Procedure for Product Specification Recommendation Optimization with LDA-LightGBM and QFD

**Tzu-Chien Wang *** , **Ruey-Shan Guo and Chialin Chen**

Department of Business Administration, National Taiwan University, Taipei City 106, Taiwan; rsguo@ntu.edu.tw (R.-S.G.); cchen026@ntu.edu.tw (C.C.)
* Correspondence: d08741009@ntu.edu.tw

**Abstract:** E-commerce and social media have become increasingly essential and influential for sustainable business growth, particularly due to the COVID-19 pandemic, which has permanently altered the business landscape. The vast amount of consumer data available online holds significant potential and value. The strategic utilization of this information can expedite the research and development of new products, leading to shorter product cycles and increased innovation. This study explores the effectiveness of employing the latent Dirichlet allocation (LDA) method and various deep learning technologies to predict Amazon consumer ratings. We propose a product service system that utilizes natural language analyses of online sales data and user reviews, enabling industries to quickly identify and respond to market demands. We present a data-driven procedure for the customer-to-manufacturer (C2M) business model, specifically focusing on sustainable data-driven business models based on knowledge and innovation management. This procedure analyzes user comments on online shopping platforms to match product requirements and features, optimize product values, and address issues related to product specifications and new product development planning. The results of the business verification demonstrate that this procedure accurately evaluates product specifications under different demands, facilitates effective product planning, and enhances research and development decision making. This approach, based on sustainable data-driven business models and knowledge and innovation management, expands market opportunities for the sector and improves overall production efficiency, starting from the research and development stage.

**Keywords:** product service system; latent Dirichlet allocation; consumer-to-manufacturer; sustainability; gradient-boosting decision tree methods



## 1. Introduction

The emerging C2M (customer-to-manufacturer) and D2C (direct-to-customer) business models have initiated a movement towards e-commerce, especially during the COVID-19 pandemic. Direct-to-consumer (D2C) e-commerce sales in the U.S. are expected to reach USD 175 billion by the end of 2023, with this success attributed to the transformation of the role of original equipment manufacturers (OEMs). The use of consumer data plays a vital role in manufacturing, research, and development (Mak et al., 2021 [1]; Hinterhuber, 2022 [2]). For example, in the case of the Alibaba Group in China, consumer data are often used to address issues related to design and production, with user comments being quickly assessed to align products with their preferences. The C2M model strengthens the connection between R&D and the market, reduces the cost of information exchange between engineering and marketing departments, accelerates development, innovates product features, and enhances the production process's flexibility to meet ever-changing demands. Moreover, it opens up new opportunities for digital transformation in the manufacturing industry (see Mak, H. Y., and Max Shen, Z. J. (2021) [1] for the details of the C2M framework).

With the aim of achieving sustainable business goals, our research advocates a data-driven approach for the R&D processes of small and medium-sized manufacturers. Utilizing a data-driven C2M model, we emphasize the comprehension of consumer needs and desired product specifications through online data analysis. This methodology accelerates the research direction and enhances production efficiency (Relich, M., 2023 [3]). Furthermore, our product specification recommendation system is pivotal for the manufacturing industry. It identifies coveted product features, amplifying the potential for successful sales. This system draws from the existing literature, addressing data collection, product feature alignment, and integration into management systems. Data-driven decision making empowers businesses to adeptly respond to market demands and foster sustainability.

We utilize a natural language processing (NLP) system founded on a "customer demand–product specification–value" framework to extract keywords and sentiments from user reviews. This guides customer demands and product specifications using latent Dirichlet allocation (LDA) (Wang et al., 2020 [4]; Mohammad et al., 2021 [5]). The nexus between consumer needs and product specifications is realized via gradient-boosting decision tree methods (Patnaik et al., 2021 [6]). The R&D process evolves due to quality function deployment (QFD), aligning product specs with customer needs (Balakrishnan et al., 1996 [7]; Altiparmak et al., 2009 [8]; Wang et al., 2012 [9]). Our data-driven approach seamlessly integrates into the product service system (PSS) via QFD, bolstering responses to shifting market demands and fostering customer loyalty (Guo et al., 2021 [10]; Yang et al., 2021 [11]). Manufacturers can center their efforts on sustainable solutions, encompassing eco-friendly materials and production methods. Our study zeroes in on the transition of Taiwanese manufacturers from OEM to OBM, utilizing gradient-boosting decision tree methods with QFD for the C2M model. Through the amalgamation of market insights with R&D, the manufacturing industry can craft sought-after, sustainable products and elevate sales on Amazon. This methodology is relevant to diverse manufacturing sectors.

This study contributes to the field of sustainable data-driven business models based on knowledge and innovation management, specifically in contrast to prior research on product service systems (PSSs). We propose an innovative approach called "integrated latent Dirichlet allocation and gradient-boosting decision tree methods", which focuses on new product development trends and data-driven decision making. By utilizing real-world manufacturing scenarios and product data, we introduce a C2M data-driven framework for precise customer rating predictions and targeted product specifications. This approach differs from previous studies as we present an empirical model that combines gradient-boosting machine learning with latent Dirichlet allocation topic analysis, leading to improved predictive accuracy and generalization. Additionally, the integration of the QFD framework enhances product development decision making, aiding OEM enterprises in defining specifications, streamlining development, and aligning products with market needs. These insights provide valuable practical guidance for data-driven sustainable business model research.

Despite these contributions, this research is subject to some limitations. First, the definition of key terms such as customer needs, product functions, and product specifications in the subject analysis relies on the subjective judgment of the product manager, necessitating input from individuals with industry expertise. Second, the construction of the "customer demand–product specification–value" hierarchy heavily depends on the results derived from the feature matrix of thematic analysis. Third, the scope of this research is confined to a case study within the B2C manufacturing sector, specifically focusing on end products. Thus, the applicability of our case study should be broadened to encompass a wider range of industries. Lastly, sustainable operations should encompass multiple objectives beyond mere corporate profitability. However, this study did not account for lifecycle indicators, which entail assessing short- and long-term impacts on the environment and human health. Addressing this gap is imperative for future research endeavors.

The paper is structured as follows: Section 2 reviews the materials and methodologies related to our study, providing the analytical framework. Next, Section 3 systematizes

the architecture, detailing the proposed C2M data-driven product–service system, and Section 4 explains the results of the analysis. In Section 5, we illustrate the application of this process in real-world scenarios and present the main findings with practical examples. Section 5 summarizes the conclusions of this paper.

## 2. Materials and Methods

### 2.1. Conceptualizing the C2M Data Model

The customer-to-manufacturer (C2M) model has transformed manufacturing, directly linking customers and manufacturers for the provision of personalized products. This eliminates steps such as inventory and distribution, yielding cost-effective, high-quality goods with direct customer involvement (Fan et al., 2021 [12]). Retailers use mobile apps to gather preferences, inform production, and create desired items, shaping customer perceptions through brand experience. A platform coordinator customizes product specs based on demand, enhancing efficiency and innovation through data analysis and creating a competitive edge. JD.com's C2M model adopts a pull production strategy, optimizing scheduling via data analysis (following Amazon's lead). Investments in data analysis, including forecasting and pricing models, yield product differentiation and brand distinctiveness. Integrating R&D, marketing, and data analysis empowers companies to base development, production, and sales on consumer data. Feedback analysis in physical and virtual channels fosters non-cost-based competitive advantages. Despite challenges in data integration and analysis, incorporating big data and product lifecycle systems enhances innovation (Tao et al., 2018 [13]). Knowledge management extracts value from accumulated product data, enabling quantitative analyses and a knowledge integrated production system for development and inference. A diagram representing the conceptual C2M data model for this study is given in Figure 1.

**Figure 1.** The conceptual C2M data model.

### 2.2. Proposed Framework

Figure 2 illustrates the framework adopted in this study. The structured data collected from customer purchases and product reviews were integrated, including the daily sales amount of the product obtained from the Amazon data provider, the Amazon product categories, and customer ratings. Additionally, unstructured data encompassing intrinsic attributes and extrinsic attribute measurements in the product attribution, consumer's hierarchy of needs, and product benefits were also incorporated. A machine learning model was employed to predict the product satisfaction of Amazon customers. Finally, six different metrics were utilized to evaluate the predictive performance of the model.

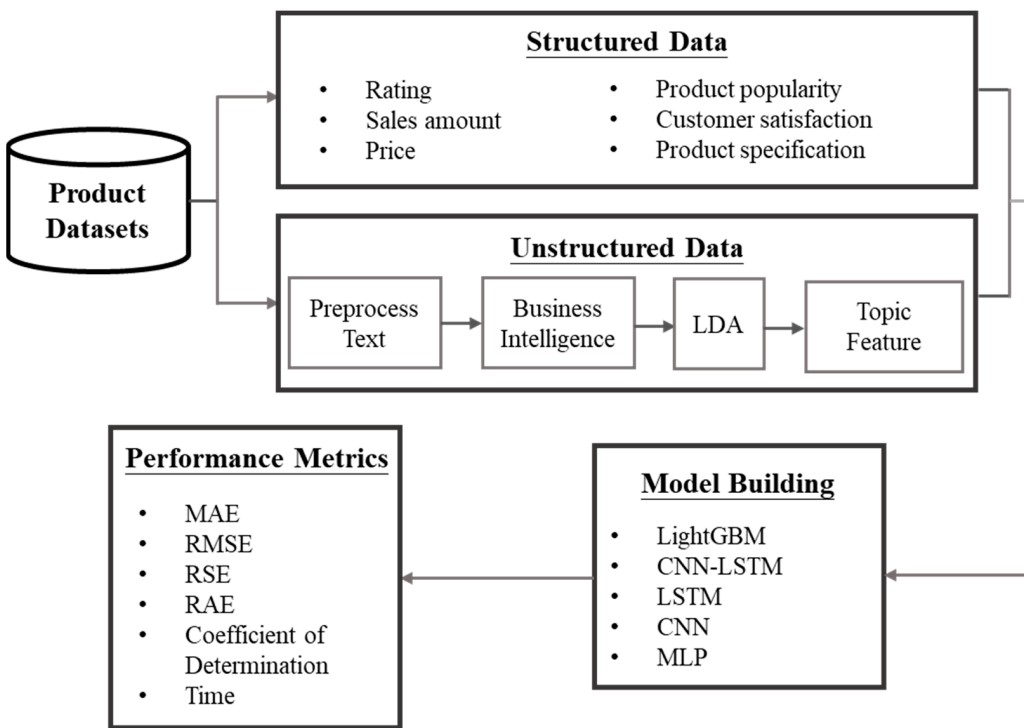

**Figure 2.** Research scheme.

*2.3. Case Study*

By employing the C2M (customer-to-manufacturer) model and a data-driven approach, this study aims to contribute to the field of sustainable data-driven business models based on knowledge and innovation management. It enables a comprehensive analysis by integrating structured and unstructured data, providing valuable insights into customer satisfaction predictions in the context of Amazon. The evaluation metrics employed ensure a rigorous assessment of the model's performance, offering a reliable basis for decision making in the pursuit of sustainable business growth. In the context of this study, the C2M model is applied to a Taiwanese electronics manufacturing firm specializing in power supply and transformer systems, using a B2B (business-to-business) approach. The company has made significant strides in terms of charger technology, incorporating environmentally friendly Gallium nitride (GaN) as a core component, resulting in ultra-small power cubes with high charging efficiency. The company's commitment to sustainability is evident through its motto "One for All", emphasizing lightweight, efficient, and universally compatible chargers designed to reduce environmental impacts and cater to consumers' daily needs. The data-driven approach extends to the development of sustainable solutions, where the company seeks environmentally friendly materials and production methods. Sustainability standards and guidelines are introduced to assess new products' sustainability performance and impact, considering consumer concerns about sustainability and social responsibility in market demand assessments. This process ensures that the product development aligns with sustainability goals while meeting consumer preferences. With real-time data collected from various regions, the company proactively responds to rapidly changing market demands, adapting its product development direction and marketing strategy accordingly. This approach not only enhances business performance on platforms like Amazon but also contributes to a more sustainable and responsive manufacturing process. In conclusion, this study emphasizes the importance of data-driven business models in promoting sustainable development. By incorporating sustainable practices and leveraging data analysis and decision-making, the company can develop preferred products that resonate with consumers while reducing environmental impacts, ultimately contributing to a more sustainable and responsible approach to production and consumption.

This prototype system achieves a modular framework comprising the product data application programming interface (API) stream, customer need analysis, and automatic product specification recommendations. Product categories in both PDM and PLM are integrated into the crawling robotic process automation (RPA) platform and Amazon's data provider platform via API services. The system suggests product specification candidates aligned with each customer's needs while also providing comprehensive explanations for each specification. By doing so, product managers can effectively circumvent the time-consuming processes of market research and product planning, thereby uncovering new potential products.

In our research, the data were obtained from Amazon's database. The utilized Amazon data consisted of information for 76 product items, such as the product specifications, size, weight, and feature values provided by Amazon vendors. These categories and commodities were selected for tracking purposes by the manufacturing industry. There were 23 categories of product specifications, including size, weight, features, USB Ports, power output, output wattage, fast-charging technology, certification, advertisement, cable, charging speed, compatible devices, connection, design, durability and quality, fast-charging technology, GaN, out wattage, price, safety and protection, USB Ports, and warranty. Furthermore, the researchers generated the following data per customer per purchase: 3,492,632 product review observation records, 76 product items, and data records from January 2021 to August 2022. Additionally, the researchers reviewed previously proposed dimensions for measuring product attributes and cross-referenced them with evaluation items on the Amazon website to confirm their alignment with the actual product measurement dimensions (see Table 1 for details). Structured data sales data, scoring data, and traffic data are provided by Amazon's data provider, DataHawk. This study subscribes to the product data and the data are provided with a real data error of plus or minus 10% to protect the personal data rights of merchants (DataHawk URL: https://datahawk.co/ accessed on 30 September 2022). This study comprised a compilation of products relevant to the manufacturing industry, accompanied by a roster of their respective product introduction pages, as outlined in Table 2. Amazon utilizes the designated UPC of a product to generate its proprietary ASIN (Amazon Standard Identification Number), which is tailored to that particular item. ASINs serve as distinctive codes ascribed by Amazon to items within their catalog, facilitating streamlined searching and listing functionalities on their platform. The confirmation of data sources prompted the establishment of API streams to collect customer comments and feedback. Amazon's review data, procured through rapidAPI, Datahawk, and Power Automate, underwent a three-step transformation process from unstructured to structured data. Initially stored in .json files, the API data underwent cleaning and conversion to structured .csv files. This standardized data were subsequently loaded into a cloud database.

**Table 1.** Reference table.

| Product Attribution | Reference |
| :---: | :---: |
| Price | Dodds, W. B., Monroe, K. B., and Grewal, D. (1991) [14] |
| Brand | Richardson, P. S. and Dick, A. S. (1994) [15] |
| Durability | Rao, A. R. and Monroe, K. B. (1988) [16] |
| Quality | Lee, M. and Lou, J. (1996) [17] |
| Warranty | Purohit, D. and Srivastava, J. (2001) [18] |
| Advertising | Boulding, W. and Kirmani, A. (1993) [19] |
| Compatibility | Katz, M. L. and Shapiro, C. (1986) [20] |

**Table 2.** Partial list of data sources for the manufacturing industry.

| ASIN | Category | Reviews | Price |
|------|----------|---------|-------|
| B09MV3M4GL | Cell phones and accessories | 11,899 | 51.62 |
| B07PZSXL9J | Cell phone wall chargers | 8017 | 39.99 |
| B01IUTIUEA | Cell phone wall chargers | 5660 | 39.99 |
| B00P933OJC | Cell phone wall chargers | 3765 | 30.99 |
| B07D64QLQ1 | Laptop chargers and adapters | 3754 | 25.99 |
| B097PTBB5V | Laptop chargers and adapters | 3744 | 29.99 |
| B07ZCGYP27 | Cell phone wall chargers | 2928 | 54.99 |
| B08KTG9L3H | Chargers and adapters | 2444 | 49.99 |
| B07VSMK849 | Cell phone wall chargers | 2150 | 24.53 |
| B07DFGXLY4 | Cell phone wall chargers | 2120 | 119.99 |
| B08YJLMQGD | Cell phone wall chargers | 2047 | 49.99 |

The C2M model's implementation approach, pivotal results, and significant findings for this case study are succinctly outlined below:

*Terminal Market Data Acquisition Mode:* The implementation of this system instigates a transformative enhancement to the existing product database. The transition from traditional internal database analysis to a holistic internal and external data model proves pivotal. By orchestrating the automatic collection of external product data and the subsequent generation of structured information via the ETL process, the system ensures expedited access to terminal customer insights. Consequently, the arduous task of market data retrieval is substantially abridged.

*Customer Need Analysis Findings:* When a product manager sets out to craft innovative offerings, the system rapidly furnishes real-time insights, encompassing customer demands, product attributes, inherent product functionalities, and the most sought-after items in the current market. By categorizing a spectrum of product requirements, the system equips product managers with the ability to discern key elements and assign them relevant significance rankings. The availability of a topic analytical model expedites the intricacies of market research, thereby considerably aiding in the meticulous realm of product planning.

*Product Specification Recommendation Model:* An innovative facet of this system involves the establishment of a robust foundation in terms of product specification knowledge, complemented by the framework of quality function deployment (QFD). This is further reinforced by a self-learning mechanism. The system autonomously elucidates intrinsic relationships between product functionalities and specifications. Manual adaptations seamlessly integrate into the knowledge base, subsequently fortifying the training model via the self-learning mechanism. This meticulous self-study mechanism distinctly elevates the reliability quotient within the realm of burgeoning interdisciplinary domains.

This AI-based system not only enhances the efficiency of analyzing extensive text data but also seamlessly integrates and evaluates customer needs and product specifications. This accelerates various dimensions of product development, including ideation, market research, and product specification assessment. C2M constitutes a significant application within the realm of smart manufacturing, where the fusion of big data and artificial intelligence plays a pivotal role. The integrated algorithm and prototype system possess the ability to extract semantic insights from e-commerce review texts, thus propelling intelligent product development. For a comprehensive understanding of the C2M model's implementation approach, please consult Sections 2.6 and 2.7 for more detailed information.

### 2.4. Latent Dirichlet Allocation and Customer Need Variables

The integration of topic methods and machine learning models proves effective in extracting text features and enhancing predictive model accuracy. For instance, Lijuan Huang et al. (2019) [21] introduced a topic model based on the Supply-Chain Operations Reference (SCOR) framework, which merged a sentiment model to analyze online text reviews and forecast sales performance. Their study revealed that, with certain topic

probabilities prevailing, the sentiment within review texts exerted a substantial influence on sales. This influence amplified forecasting precision, particularly in the realm of online sales, surpassing alternative methods (Huang, L., Dou, Z., Hu, Y., and Huang, R., 2019) [21]. Moreover, within the e-commerce landscape, where product brand construction and sales stimulation are paramount, the amalgamation of machine learning and topic analysis methods facilitates the extraction and filtration of emotional information from online reviews. This culmination optimizes nonlinear regression models, predicting product sales with heightened accuracy. This model not only observes actual sales for precise predictions but also adeptly discerns market trends and values (Huang, L., Dou, Z., Hu, Y., and Huang, R., 2019) [22].

This study investigates the effectiveness of using the latent Dirichlet allocation (LDA) method to analyze customer need variables and product attributes. LDA is an unsupervised topic modeling algorithm that extracts topics from a corpus of text. It assigns a distribution of topics to each document in the collection and a distribution of words to each topic. By iteratively analyzing the words in the documents, LDA identifies the most representative words for each topic, enabling the discovery of latent themes and semantic patterns within the text. LDA assumes that all documents share the same topics to varying degrees. Using the Gibbs sampling algorithm, LDA infers the underlying topics in a set of documents and generates a topic distribution for each document (Blei, D. M., Ng, A. Y., and Jordan, M. I., 2003; Blei, D. M. and Lafferty, J. D., 2007) [23,24]. LDA adopts a typical "bag of words" approach, treating each text as a vector of vocabulary frequencies and as an amalgamation of various vocabulary groups. Each of these vocabulary clusters corresponds to a distinct topic, with the extraction of text topics happening independently of vocabulary order and interrelation. Typically, LDA constructs its model for generating topics through a sequence of steps: (1) the selection of a topic from the array of topics within a text; (2) the choice of a vocabulary from the topic-related vocabulary list; and (3) the repetition of this process until all vocabularies in the text are accounted for. Given that $i$ denotes the document number, $K$ signifies the number of topics, $Zij$ indicates the frequency of vocabulary $j$ appearing across different topics in Document $i$ (with $Zij$ following a multinomial distribution), $\theta i$ represents the probability of each of the $k$ topics occurring in document $i$ (with the Dirichlet distribution, having a hyperparameter $\alpha$, serving as the prior distribution), $Wij$ signifies the nth vocabulary in document $i$ (following a multinomial distribution), and $\Phi k$ denotes the probability of each vocabulary within the kth topic occurring (with the Dirichlet distribution, having a hyperparameter $\beta$, serving as the prior distribution), an overarching LDA framework, as illustrated in Figure 3, can be established.

The LDA model assumes that words are generated by topics, and topics in the text are infinitely exchangeable. De Finetti's theorem states that, for a set of random samples that satisfy exchangeability, their joint distribution remains unchanged regardless of the permutation order of random variables. Moreover, an infinite sequence of exchangeable random samples is equivalent to sampling random parameters from a prior distribution, thus allowing for the resampling of independent and identically distributed random variables. Therefore, given prior parameters α and β, the joint distribution of topic distribution probabilities θi, topic set Zij, and word set Wij can be expressed as:

$$P(\theta i, Wij, Zij, \Phi k | \alpha, \beta) = P(\theta | \alpha) \prod_{n=1}^{N} P(Zij | \theta) P(Wij | Zij, \beta) \tag{1}$$

To operate the LDA topic model, it is also necessary to compute the posterior distribution of latent variables:

$$P(\theta i, Zij | Wij, \alpha, \beta) = \frac{P(\theta i, Wij, Zij | \alpha, \beta)}{P(Wij | \alpha, \beta)} \tag{2}$$

In addition to obtaining the document distribution, topic distribution, and word distribution of the LDA model through variational inference and the EM algorithm (Blei, Ng, and Jordan, 2003) [23], Gibbs sampling is a commonly used method. Gibbs sampling relies on the principles of Markov Chain Monte Carlo (MCMC). It assumes that, in a

two-dimensional space of probability distribution $P(x, y)$, while fixing one dimension's X coordinate, the conditional probabilities can be calculated as follows:

$$P(x_1, y_1)P(y_2|x_1) = P(x_1) \ P(y_1|x_1)P(y_2|x_1) \tag{3}$$

$$P(x_1, y_1)P(y_1|x_1) = P(x_1) \ P(y_2|x_1)P(y_1|x_1) \tag{4}$$

This leads to:

$$P(x_1, y_1)P(y_2|x_1) = P(x_1, y_2) \ P(y_1|x_1) \tag{5}$$

When applied to high-dimensional spaces, this method can be used to iteratively solve the posterior distribution of the LDA model. To calculate $p(w|\alpha, \beta)$, all other variables in each iteration can be fixed and a single latent variable sampled, thus obtaining the posterior distribution.

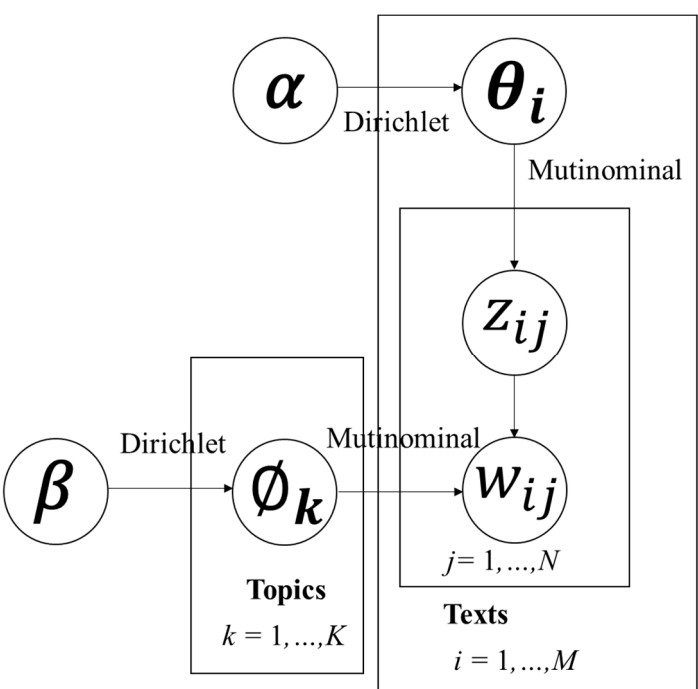

**Figure 3.** LDA model framework.

LDA is the preferred method for feature extraction in assessing the impact of product reviews and user ratings. It automatically identifies underlying topic structures within text data, enabling a profound understanding of the text's meanings and themes. Organizing words into topics, LDA unveils correlations and hidden insights, enriching machine learning models with intricate features and enhancing analytical and predictive prowess. In product review analysis, LDA exposes diverse topic traits, aiding machine learning in understanding consumer evaluations. These topic structures also shed light on machine learning predictions. We can discern how specific topics' demand and attributes affect projections. For instance, if a model predicts a high rating for a review, we can elaborate on the topic structure and product attributes behind this. Integrating LDA and machine learning enhances predictive capabilities and the comprehensibility of the results, strengthening overall interpretability.

The decision to select five themes was driven by the need to achieve a balanced and insightful analysis. The risk of using too few themes lay in the potential oversimplification of the findings, while an excessive number of themes could lead to fragmented insights. Moreover, considerations related to computational resources, efficiency, and model interpretability exerted a significant influence. The selection of an appropriate number

aimed to harmonize the depth of analysis with practical feasibility (Abdelrazek, A., Eid, Y., Gawish, E., Medhat, W., and Hassan, A., 2022) [25].

The steps are organized as follows:

1. Review of Product Variables: The study commenced with a comprehensive literature review (Table 1), aimed at identifying the dimensions and variables that influence customer perceptions of product utility. This phase served as the foundation, illuminating multifaceted factors that collectively shape product evaluations.

2. Discussions with the Product Manager: Extensive dialogues with the product manager of the case company enriched the study with valuable industry insights. These discussions adeptly uncovered practical viewpoints on pivotal product specification classifications, specific product certifications, technological considerations, and prevailing consumer expectations.

3. Determining the Optimal Number of Topics: The methodology involved utilizing visualization techniques to interpret the outcomes and obtaining feedback from the product manager to assess the effectiveness of the derived topics. As a result, a set of five principal themes was selected: technical and performance assessment (Certification), functional requirements and improvement suggestions (Charging Speed), value and price evaluation (Price), product quality and durability (Quality and Durability), user experience and brand perception (Brand). This selection yields a holistic representation of the identified dimensions.

Based on the research findings, this study utilizes the LDA model to analyze customer reviews of charger competitors on Amazon and identifies five themes. To gain a better understanding of these themes, the researchers identified keywords associated with each theme using input from the case company's product manager and literature on product attributes. This approach aims to explore the managerial implications of each topic.

The first topic encompasses keywords related to charger brand names and phone device compatibility, such as "Samsung", "Apple", "Anker", and "ZMI". This topic is named "Brand Awareness", based on the marketing literature, highlighting its connection to brand recognition and context.

The second topic primarily includes keywords related to charging power and speed, such as "charging efficiency," "fast charge", "speedy", and "charging power". It is named "Charging Speed" to emphasize users' perceptions of charger performance and charging times.

The third topic revolves around keywords related to purchase intent and price, such as "cheap", "affordable", "costly", and "budget-friendly". it is named "Price" based on literature examining the influence of purchase intention, focusing on content related to purchase intent and pricing.

The fourth topic consists of keywords related to charger certification and technical specifications, such as "Live Mall", "PD 3.0", "PDO", and "TUV certification". This topic is named "Certification" to underscore its relevance to product certification and technical specifications.

Finally, the fifth topic incorporates keywords related to quality, durability, and safety, such as "stopped working", "overcurrent protection", "robust noise immunity", and "spotty". It is named "Quality and Durability" to emphasize aspects of product quality, durability, safety, and protection (Table 3).

These analysis results provided researchers and R&D personnel with a deeper understanding of customer feedback on the Amazon platform, enabling them to formulate product development strategies that align with different customer needs (Qin, C., Zeng, X., Liang, S., and Zhang, K., 2023) [26].

LDA proves effective in extracting features from rating reviews; however, its inherent limitations warrant consideration. The assumption of fixed topic distributions results in inaccuracies, failing to capture evolving word–topic dynamics. LDA's capacity to grasp subtle nuances is compromised, hinging on word co-occurrence while neglecting essential contextual word order. The intricate task of topic selection underscores the challenge.

Integrating LDA with machine learning requires meticulous validation to mitigate the risk of overfitting. To tackle contextual intricacies, a promising avenue involves augmenting LDA with sequential processing machine learning algorithms, thereby enhancing its contextual comprehension and analytical performance. This augmentation facilitates a more comprehensive understanding of the underlying dynamics in review analysis.

**Table 3.** Topics and keywords for the dataset.

| | Topic 1:<br>Brand | Topic 2:<br>Charging Speed | Topic 3:<br>Price | Topic 4:<br>Certification | Topic 5:<br>Quality and Durability |
|---|---|---|---|---|---|
| 1 | Anker | Charged fast | Cheap | TUV certification | Break your phone |
| 2 | Samsung | Charges fast | Expensive | IEC 62368-1 | Robust noise immunity |
| 3 | Nekteck | Charging efficiency | Pricy | USB-IF | Overcurrent protection |
| 4 | Apple | Charging speed | Affordable | UL certification | Died |
| 5 | Spigen | Charging time | Cost-effective | UE | Quit working |
| 6 | Innergie | Consistent power | Reasonable | FCC | Spotty |
| 7 | Syncwire | Efficiently | Costly | SGS NA listed | Stop charging |
| 8 | Nekmit | Enough power | Inexpensive | DOE Level VI | Stopped working |
| 9 | ZMI | Fast | Awesome price | CE | Well built |
| 10 | TECKNET | Fast charge | Cheaper | ETL | Temperature |
| 11 | Baseus | Have a fast charge | Reasonably priced | RoHS | Worked flawlessly |
| 12 | UGREEN | High-speed charging | Great value for money | PSE Certificate | Pitched sound |
| 13 | Hyphen-X | Lose power | Great price | PD 3.0 | Sketchy |
| 14 | AOHI | Max output | Good price | PDO | Terrific |
| 15 | Ixcv | Powerful | Pricey | PPS | Charge intermittently |

### 2.5. Machine Learning and Deep Learning

This study investigates the effectiveness of employing the latent Dirichlet allocation (LDA) model, along with various regression methods, to predict Amazon consumer ratings. However, traditional statistical regression analysis may not be suitable for analyzing review data due to violations of assumptions such as non-linearity, heteroscedasticity, and non-normality. Moreover, the qualitative aspects of review data present challenges to conventional regression frameworks, making direct application to censored data risky. To address these challenges, previous studies highlight the significance of neural network methods, particularly light gradient-boosting machines (LightGBMs), long short-term memory (LSTM), and convolutional neural networks (CNN), as they are better suited to extracting insights from review datasets.

In this study, our primary objective is to enhance the neural regression model by integrating LDA. The dataset is partitioned into two segments: one designated for model training, and the other for testing purposes. We utilized four widely employed machine learning algorithms to formulate predictive models for review data: latent Dirichlet allocation (LDA), a light gradient-boosting machine (LightGBM), long short-term memory (LSTM), and CNN with CNN-LSTMs. Our comprehensive model comparison endeavors to pinpoint the optimal predictive model for Amazon consumer ratings. All data mining tasks conducted within this study were executed using the Python programming language. Moreover, this research deliberately selected the integration of sequential processing machine learning techniques to account for the nuanced nature of review data.

The following section provides an in-depth introduction to and comparison of the four models.

- Long short-term memory (LSTM): LSTM is a type of recurrent neural network (RNN) specifically designed for handling sequential data, such as text. It effectively tackles the vanishing gradient problem by incorporating memory cells that can retain information over long sequences. In the context of processing review data, LSTM proves valuable in analyzing the sequential nature of reviews, capturing dependencies between words or phrases to comprehend the underlying sentiment or meaning. By learning patterns

from review data, LSTM models can predict and classify reviews based on sentiment or other relevant attributes (Hochreiter, S. and Schmidhuber, J., 1997) [27].

- Convolutional neural networks (CNNs): a CNN is a deep learning architecture widely used in computer vision tasks, but it can also be adapted for text data. In the context of review data, CNN can extract pertinent features from text by applying filters or convolutions over the input. These filters can capture patterns or word combinations that are indicative of the sentiment or other characteristics of the reviews. Utilizing multiple layers of convolutions and pooling operations, CNN models can learn hierarchical representations of text data, enabling them to predict or classify reviews based on these learned features (Kim, Y., 2014) [28].

- CNN-LSTM: CNN-LSTM is a hybrid model that combines the strengths of a CNN and LSTM. It is particularly effective in handling sequential data with both spatial and temporal dependencies, as found in text data. In the context of review data, a CNN-LSTM model uses CNN layers to extract features from the text and then passes these features to LSTM layers to capture the sequential dependencies. This combination allows the model to grasp both local patterns in the text and long-term dependencies, improving its ability to understand the sentiment or other characteristics of the reviews (Onan, A., 2021) [29].

- LightGBM (light gradient-boosting machine): a LightGBM is a gradient-boosting framework that employs decision trees as base learners. It is well-regarded for its efficiency and capacity to handle large datasets. LightGBM functions by iteratively adding decision trees trained to correct mistakes made by previous trees. It utilizes gradient-based one-sided sampling to select the most informative instances for building trees, enhancing both speed and memory usage. In the context of review data, LightGBM can be utilized for regression tasks to predict numerical values, such as the number of reviews or ratings. Additionally, it can handle categorical features and is commonly employed for feature selection, classification, and regression tasks across various domains (Ke, G., Meng, Q., Finley, T., Wang, T., Chen, W., Ma, W., and Liu, T. Y., 2017) [30].

- After evaluating diverse deep learning sentiment models, the integration of the gradient-boosting decision tree (LightGBM) stands out due to its unique advantages. LSTM efficiently handles sequential data and dependencies, capturing nuanced sentiment in reviews. CNN employs convolutional filters for pivotal feature extraction in sentiment classification. CNN-LSTM hybrids adeptly manage spatial and temporal dependencies in sequential text data. Unlike other models that focus on the LDA context, LightGBM's strength lies in managing data imbalances. It excels in addressing data skewness through gradient-based one-sided sampling, which is crucial when handling reviews with uneven sentiment distribution. This strengthens sentiment analysis by ensuring a balanced approach, making LightGBM an apt choice to enhance sentiment classification precision and robustness.

- Below is an introduction to the LightGBM objective function, loss function, and regularization term.

**Objective Function:**

The objective function of LightGBM combines the loss function and regularization term and can be represented as:

$$Objective(f) = \sum_{\{i=1\}}^{\{n\}} L(y_i, f(x_i)) + \sum_{\{k=1\}}^{\{K\}} \Omega(f_k) \tag{6}$$

where $y_i$ represents the true label of the i-th sample, $f(x_i)$ represents the model's prediction for the i-th sample, and $f_k$ represents the k-th tree in the ensemble model. The function $L$ denotes the loss function, and $\Omega$ represents the regularization term.

**Loss Function:**

The choice of loss function depends on the specific task. For regression problems, common options include:

$$\text{Mean Squared Error (MSE): } L(y, \hat{y}) = (y - \hat{y})^2 \tag{7}$$

$$\text{Mean Absolute Error (MAE): } L(y, \hat{y}) = |y - \hat{y}| \tag{8}$$

For a detailed introduction to the loss function indicators, please refer to the research by Dou, Z., Sun, Y., Zhu, J., and Zhou, Z. (2023) [31].

**Regularization Term:**

LightGBM supports various regularization methods, such as *L1* regularization (Lasso) and *L2* regularization (Ridge). The specific form of the regularization term $\Omega$ depends on the chosen regularization method and is added to the objective function to prevent overfitting.

In this study, we employed a 10-fold cross-validation approach for training and testing. Specifically, 80% of the data were utilized for model training, while the remaining 20% were reserved for model testing. Subsequently, we conducted comprehensive model comparisons to assess the performance.

*2.6. Quality Function Deployment*

Based on the current literature on product development and product quality evaluation methods, product prototypes are typically evaluated using product specification scale measurement, customer demand surveys, and the Net Promoter Score (NPS) to guide product development. Similarly, the assessment of manufacturing service quality often involves qualitative interviews, SERVQUAL, QFD (quality function deployment), TRIZ (the theory of inventive problem solving), and other approaches. However, despite the availability of various analysis methods, the most critical challenge faced by enterprises is effectively collecting instant feedback from customers to inform product design by the R&D department. While QFD can integrate information and benefit cross-functional teams, some scholars have recently adopted an integrated product development approach, combining TRIZ, QFD, and service design blueprint tools to introduce a service-oriented integrated solution (Wang, Y. H., Lee, C. H., and Trappey, A. J., 2017) [32]. Nonetheless, these QFD-based approaches still rely on traditional qualitative questionnaires and focus groups to capture customer needs and satisfaction dimensions, without leveraging data-driven models. In traditional B2B or B2C consumer market surveys, the collected data often lack centralization and real-time updates, containing irrelevant and simplistic information. As a result, they fail to provide companies with a comprehensive understanding of the voice of the customer (VoC). Addressing these challenges is crucial for organizations striving to achieve sustainable business growth and development.

The existing literature addresses several issues related to product development and the application of QFD. First, the list of customer needs and their importance ranking mostly rely on traditional questionnaire surveys and qualitative analysis, which may not fully capture dynamic and evolving consumer preferences. Second, there is a lack of corresponding values assigned to the product specifications based on the customer needs identified. This gap in the QFD process can hinder the ability to make data-driven product recommendations that align with user value, especially in a complex and rapidly changing consumer market.

Data-driven QFD has attracted considerable attention in recent research. One approach revolves around utilizing word frequency to analyze customer needs and objective weights, which provide the foundation for engineering parameters in subsequent product or service design (Liu, P., Zhang, K., Dong, X., and Wang, P., 2022) [33]. Another perspective suggests that machine learning methods can be employed to identify innovative services within a systems methodology, concurrently establishing new service patterns as a framework for recognizing these novel services (Ha, S., and Geum, Y., 2022) [34]. This paper integrates

the concepts from these two viewpoints to formulate a more comprehensive approach for dissecting customer needs, identifying product design parameters, and optimizing product utility. This synthesis aims to more effectively fulfill consumer needs and facilitate the creation of the most suitable products.

To address challenges and promote sustainable business models, organizations must integrate data-driven approaches into the QFD process. By leveraging sustainable data-driven methodologies, organizations can gain deeper insights into customer preferences and behavior, enabling them to prioritize product features that align with sustainability and customer satisfaction. By combining the principles of QFD with data-driven decision-making, businesses can develop products and services that not only meet customer needs but also contribute to broader sustainability goals, enabling them to better respond to the challenges of the evolving consumer landscape and contribute to building a more sustainable and resilient economy.

To address consumer demand in the market, it is crucial to integrate the outcomes of demand analysis into the product development process. This involves inputting the "customer demand" generated by LDA analysis into the Part A list (as depicted in Figure 4), which serves as the primary input for QFD value in product specification recommendation analysis. Evaluating the importance ranking entails using a machine learning method's random ranking test during market research to sort customer needs based on relative weight scores. In terms of the product's technical specifications, it is necessary to incorporate the results of customer demand analysis along with the LDA feature matrix. By setting an intercept value, higher eigenvalues of the matrix are extracted, with customer feedback relating to product design requisites being integrated into design features. The description and correlation matrix (illustrated in Figure 3) of the designed and developed product serve as inputs for the core QFD values in the product specification recommendation analysis. Correlation analysis between product technical indicators and engineering technology can be categorized into three types: strong correlation, medium correlation, and weak correlation. For instance, there is a strong correlation between volume and weight, whereas the correlation between volume and shape design is relatively weak, possibly involving a trade-off between design and technical considerations. The improvement sequence and target specification involve calculating the weighted product utility score for each engineering technology. The calculation formula is shown as Equation (4), and improvements are prioritized based on scores. A higher score signifies a more pronounced impact of the technology on customers' product evaluations. Following deliberations within the project team, specifications and goals to be achieved by various engineering technologies were established as benchmarks for product design and production.

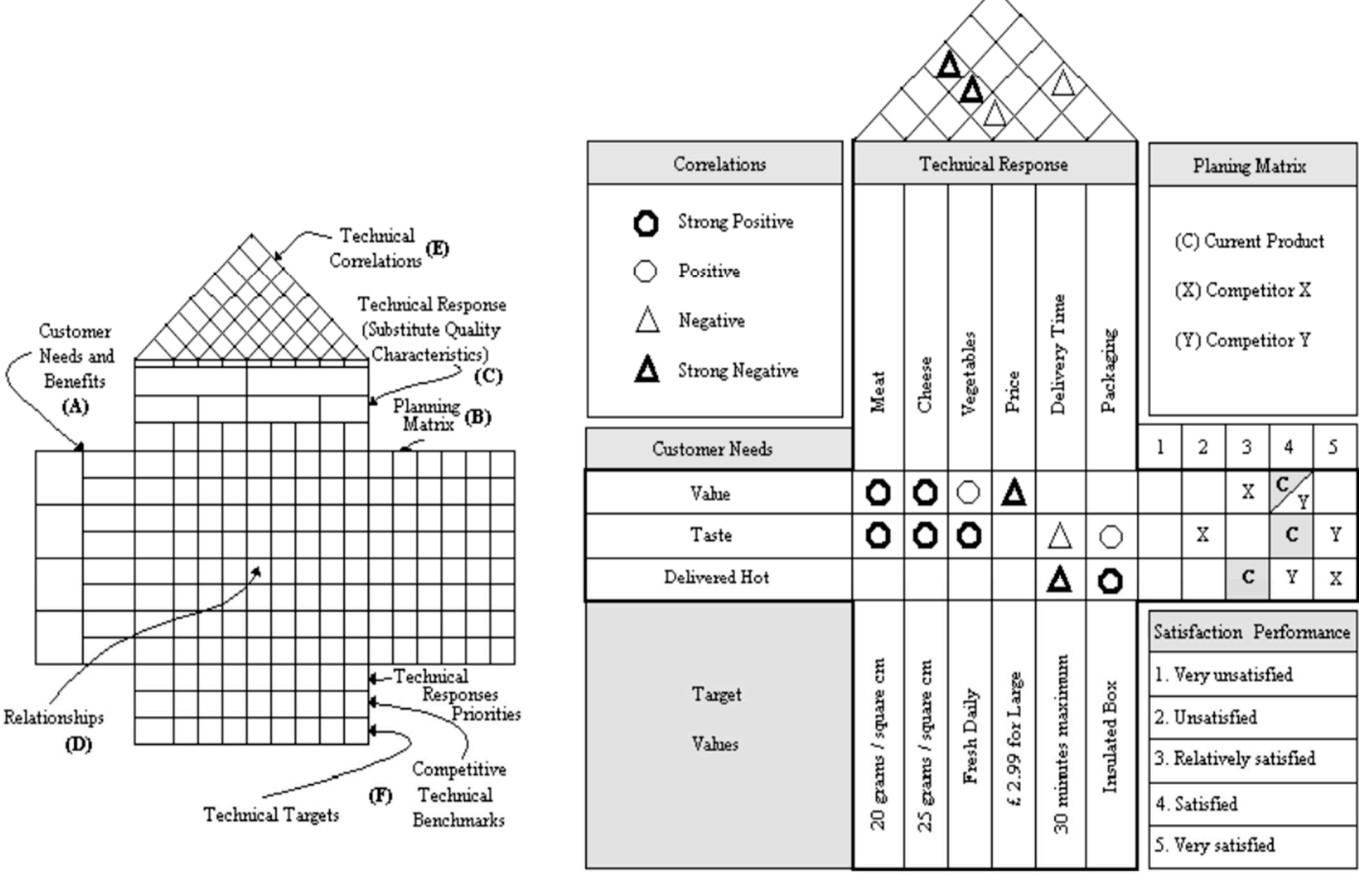

**Figure 4.** Diagram of quality function deployment (QFD). (Reprinted/adapted with permission from Ref. [35]. Copyright 2012 copyright Jaiswal, E. S.).

### 2.7. Data Preprocessing

To enhance the capabilities of QFD in rapidly determining customer needs and automating the conversion of those needs into product specifications, this study utilizes LDA (latent Dirichlet allocation) to analyze customer reviews and identify various customer needs. Each customer demand theme is then linked to corresponding product specifications within the product category. The means–ends chain theory, which explores the relationship between personal values and product choices, provides insights into consumer behavior. It categorizes consumer benefits into immediate experience and long-term value pursuit, forming an attribute–result–value (ACV) framework, where attributes refer to perceived product characteristics, consequences represent benefits, and values reflect underlying concerns pursued through consumption.

In this study, LDA is employed to connect the "customer need–product specification–value" linkage of the means–end chain theory in text mining. This analysis model compensates for potential subjective errors and effectively explores significant attribute dimensions and consumer values in the industry.

The model utilizes the star rating as the product variable, which follows a five-point Likert scale ranging from 1 to 5. The dependent variable is the customer's overall satisfaction with the product, while the independent variable is the rating based on "customer needs—product specification". To better grasp customer preferences, feature selection techniques are employed to identify crucial words that influence customer sentiment scores from Amazon online reviews. This approach enables a comprehensive exploration of customer product orientation.

Finally, the "customer need–product specification–value" framework is integrated with customer and product demand data. Gradient-boosting decision tree methods are

employed and compared with convolutional neural networks (CNNs), long short-term memory (LSTM), and CNN-LSTM technologies to determine the best prediction model. An AI-QFD (AI-assisted quality function deployment) architecture is established, utilizing this data-driven model to support the C2M (customer-to-manufacturer) business model.

### 2.8. The Procedure

In order to effectively utilize market and consumer feedback for small and medium-sized manufacturers, this study integrates the quality function deployment (QFD) framework. This integration ensures product quality and meets consumer demands by incorporating concept proposal, design, manufacturing, and service processes. By analyzing information on competing products, including specifications, sales data, and user comments, an LDA (latent Dirichlet allocation) topic model and a machine learning model are constructed to establish the relationship between user needs and product specifications. This model assists in generating recommended product designs and specifications. The research involves preprocessing market data, analyzing consumer demands, determining product fit values, recommending optimal specifications, and implementing designs within the existing product service system (PSS).

Details regarding the five steps in this study are shown below. The five system execution steps are shown in Figure 5.

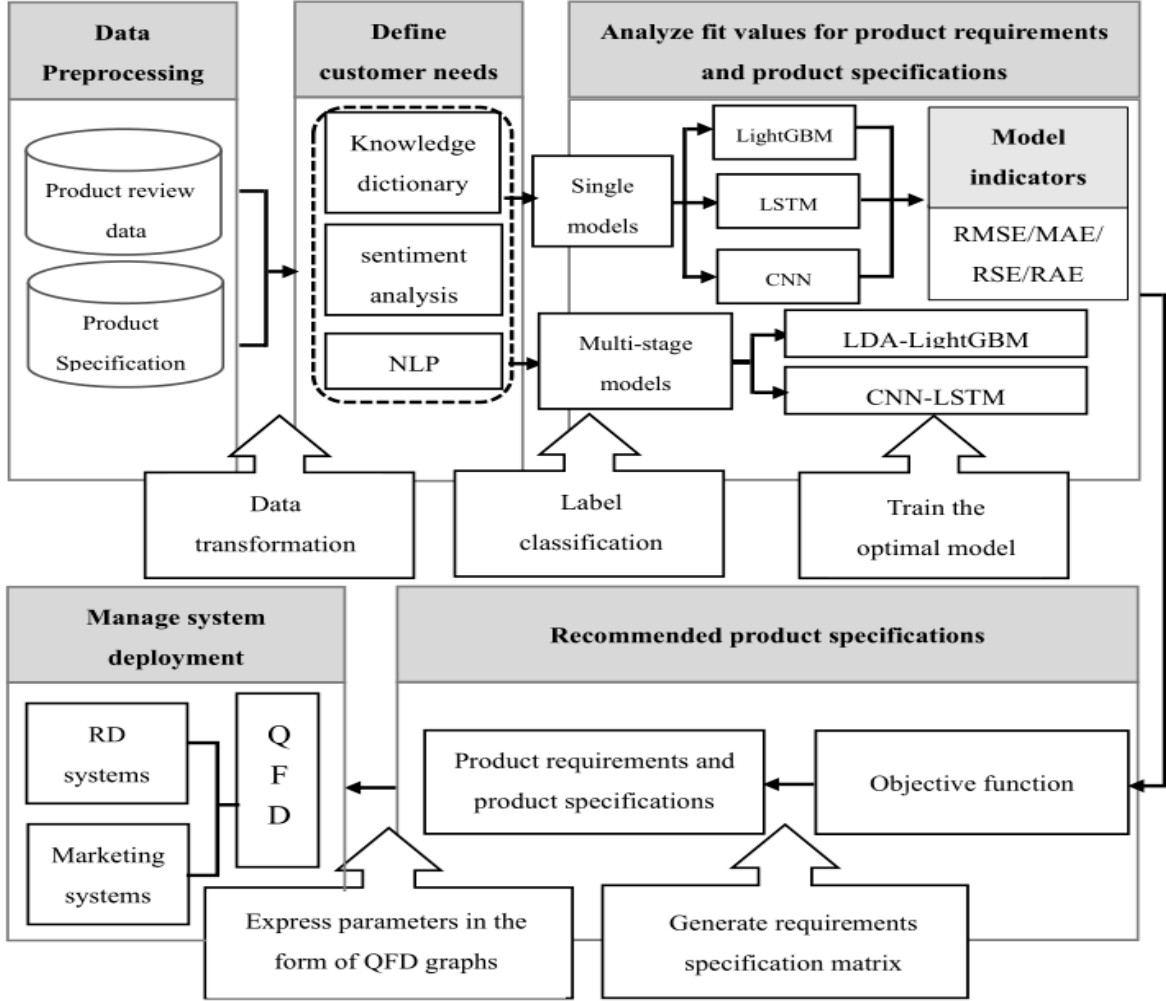

**Figure 5.** Integrative framework with a service system for the automatic identification of consumer demands.

Step 1: Pre-processing of current market data:

Using the research framework mentioned above, we worked with the example company to establish a working analysis model for the general purposes of the manufacturing industry. We collected user reviews about the charger according to the extract–transform–load (ETL) process and converted the raw data into a structured format. Then, by using the key of consumer demand–product specifications–value, we mined for the associated texts and constructed a matrix based on the user, product demand, and product specifications. Using machine learning, the system searches for matching values. After undergoing additional optimization through an adaptive heuristic algorithm, the refined values are generated as the recommended specifications. We employed the adaptive heuristic algorithm proposed by Chen, Y. et al. (2015) [36] to improve the optimization of product utility (Chaudhuri, A. and Bhattacharyya, M., 2009) [37]. This algorithm effectively captures customers' preferences for essential attributes, enabling us to enhance product variety by maximizing overall customer satisfaction and considering product specifications (Zhang, L., Chu, X., Chen, H., and Yan, B., 2019) [38]. As seen in the process, the entire flow of operation means that the output of each step is dependent on the previous input, forming an iterative data analysis loop, training the model based on sales, consumer demands, and product design requirements to formulate the optimal product specifications and marketing strategy. To analyze the mainstream online market data, we chose the API data source of the largest e-commerce platform in the United States (with gross sales of USD 367.17 billion in 2021). A schematic diagram of the structured data of product reviews is shown in Table 4.

**Table 4.** Schematic diagram of structured data of product reviews (partial).

| ASIN | Category | Review | Title | Rating |
|------|----------|--------|-------|--------|
| B07D64QLQ1 | USB-C Power adapter/charger Above 60 W | Now that most brands are switching to USB-C (apple included), it's really ridiculous that brands still charge absurd amounts of money for their own-brand charger. This will charge any 65 w device regardless of brand for a fraction of the price. Beautifully made; I plan to buy more. | Don't buy anything else | 5 |
| B07D64QLQ1 | USB-C Power adapter/charger Above 60 W | It works with the dell xps13. What I like is the compact nature of it—the small form factor is great for travel. Excellent. However, for travel and a future where you'll be in coffee shops doing work, a longer cord is needed. It looks to be about a three-foot cord which barely stretches from a floor outlet to your device. Wish it came with a 6- or 10-foot cord. | Good charger, short cord | 4 |
| B07D64QLQ1 | USB-C Power adapter/charger Above 60 W | Works great. I bought it as a spare charger. for my thinkpad, but it also charges my HP and several other USB-C devices. | Works great | 5 |

Step 2: Analysis of consumer demands:

The amount of data on the Internet is simply too gargantuan to be analyzed manually; thus, we designed a system to automatically collect and compare articles of market reviews from different locations by using feature extraction and sentiment analysis. The comments of evaluation by users, which were published online, would be mined using natural language processing when they purchased the products. Based on a semantic analysis of these texts, the system would grade users' satisfaction with the products that were currently on the market, and some key factors could be identified as the texts were disassembled and compared in terms of emotions to the specific knowledge dictionary, which was tailored with the terminology or jargon of each industry to assist not only in computational algorithms but also in the breakdown of words for natural semantic analysis. For example, the knowledge dictionary of the semiconductor industry will include the term "yellow light process", which will only be used as a whole and not as individual words. Then, the texts

were combined with sales data (such as prices, specifications, the popularity of the product inquiry, etc.) and word-of-mouth (such as the discussion of functions, product evaluation, etc.) reviews from those e-commerce platforms to be re-assembled into a useful data string. Indicative comments were found via text mining and, in combination with the knowledge dictionary and semantic analysis, words were extracted from sentences to become keywords. Then, keywords were labeled and classified, referring to "attributing result with value" as stated in the analysis theory, such as by assigning the label of "customer demand" for some keywords, which can be further categorized as "product specifications", "size", etc., based on the content. These were converted into a structured format for easier access and analysis, as well as to reduce subjective error. For example, the comment may state, "It is a great charger with high power output for laptop like XPS 15 when traveling (as indication of customer's need), that it may not quite get to full 65 w, as expected by USB-C, but it still keeps a 100 w laptop slowly charged (as technical specifications)... it must be easy to carry in briefcase (as product value) and there is no longer need to carry one huge power block or several chargers for both the phone and the laptop (as product value)... highly recommended", from which we could easily categorize and code each part of the comment based on the content.

Step 3: Identifying fit values of product requirement/specifications:

In one scenario, the LDA model generates a matrix of "customer needs–product specifications" based on individual customer requirements to assess the alignment between product variables. By analyzing customer comments, both user needs and product functionality requirements can be derived. Models are used to establish existing product specifications and predict user ratings. A trained regression model assigns weights to each independent variable, such as product demand or specification, to identify correlations between them. The process also analyzes consumer-related data to determine the direction of new product development. Since some product specifications are non-numeric features and the relationship between these data points is complex, LightGBM integration is employed to construct a regression model for rating prediction. The machine learning principle enables us to understand the relationship between product specifications and time.

In comparison with independent decision tree models in random forest, the decision tree models in GBDT (gradient-boosting decision trees) are interdependent. GBDT utilizes iterative learning, where each iteration focuses on learning the last error or residual, defined as the difference between the actual value and the last model prediction. The sum of these models yields the final prediction value. Despite their improved performance and ability to capture nonlinear relationships not captured by traditional linear methods, integrated machine learning algorithms often lack interpretability and explanation. Understanding why the machine produced a specific recommendation result or explaining the principles and mathematical knowledge behind the algorithms to industry practitioners can be challenging. However, it is crucial for industry practitioners to trust and rely on the recommendation results generated by machine learning. Hence, it is necessary to enhance the model's readability and accuracy through explainable AI.

By calculating the importance of each feature, we can comprehend the model's predicted results as the contribution of each factor. This helps to measure the influence of different factors on the prediction and provides industry practitioners with indicators for developing product design specifications. By referencing the data characteristics utilized by such a model, we can determine whether they align with the industry's heuristics or rules of thumb. Additionally, during the system development process, a deep understanding of consumer demand for products can be obtained, aiding industry players in exploring market opportunities. With a comprehension of market demand, the industry can promptly respond by developing or modifying the design specifications of new products, thereby enhancing research and development efficiency.

Step 4: Recommending optimal product specifications:

To select the best product choice, we consider technical specifications that receive high scores for both consumer demand and product requirements. By integrating the results of the QFD and the evaluated solution from model extraction, we gain a preliminary understanding of the technical specifications that are currently available on the market and provided by manufacturers. These specifications are specifically tailored to the feature preferences and needs of the target sector. To recommend optimized production specifications, we have developed mathematical calculations and an adaptive function that align with the product development process in the manufacturing industry. The relationship between customer needs and product specifications serves as the decision variable ($X_{ij}$). The adaptive function consists of the specification score ($R_{ij}$), the degree of correlation between product requirements and features ($C_i$), and the specification of each product function ($P_{kj}$). In other words, each requirement is paired with a function, and each estimated value represents a specific product specification. Despite the inclusion of penalty factors and soft constraints (which may not necessarily be followed), the result only recommends items with desirable product specifications. The function is illustrated below in Equation (4):

$$Product\ utility = \sum_i \sum_j C_i X_{ij} + \sum_i \sum_j C_i R_{ij} X_{ij} - 10^5 \left( \sum_i P_{kj} - 1 \right)^2 \tag{9}$$

Step 5: Deployment in management system:

Quality function deployment (QFD) is commonly employed in product development and marketing to align with customer preferences. By capturing the voice of the customer (VoC) and converting it into product specifications, the recommended parameters can expedite the development process and enable the delivery of desirable products or services on demand. This, in turn, aids decision making for both the marketing and technical departments. The execution framework of QFD is depicted in Figure 5. By integrating the QFD process into a company's management system, the parameters derived from analysis can be directly input into a product development system or a product marketing system to provide assistance and support in decision making.

## 3. Results

### 3.1. Prediction of Product Ratings

When considering the hyperparameters of the LightGBM model, we took several factors into account: (1) the maximum number of leaf trees, (2) the maximum number of samples in a leaf node, (3) the learning rate, and (4) the total number of constructed trees. The candidate values used to train these hyperparameters in the LightGBM model are listed in Table 5. Additionally, the table provides an example of the optimal hyperparameter values obtained through our model tuning process. Importantly, Table 5 also reveals certain patterns in the emergence of the optimal parameter range for LightGBM.

The findings presented in this table provide clear evidence that the proposed LightGBM model outperforms the other comparison models. These values indicate a smaller deviation between the predicted and actual values when the proposed model is employed. In order to assess the robustness of the proposed method, we conducted performance testing of the LDA-LightGBM model and compared it with other models using various training and testing sample sizes. The testing experiment involved adjusting the ratio of the training dataset size to the complete dataset size.

To assess the robustness of our proposed method, we conducted performance testing on the LDA-LightGBM model and compared it with other models using various training and testing sample sizes. The testing experiment involved adjusting the ratio of the training dataset size to the complete dataset size. In this section, we examined three different relative ratios. The prediction results for the five topics, obtained using the LDA-LightGBM model and the comparison models, are summarized in Table 6, with a focus on the root mean square error (RMSE) metric. Based on the observations in Table 7, the LDA-LightGBM model not only demonstrates slightly better predictive performance

compared to the other models but also showcases a significantly faster training speed. These results strongly indicate that the LDA-LightGBM approach offers substantially improved forecasting accuracy and faster training times compared to the other three approaches. The combination of LDA and LightGBM proves to be a highly effective and efficient method for the task at hand, providing more accurate predictions with reduced training times.

**Table 5.** Candidate and optimal sets of hyperparameters for the models.

| Model Names | Hyperparameter Names | Hyperparameter Values |
|---|---|---|
| LDA | LDA numbers N-gram Maximum size of N-gram dictionary | {5, 2, 20,000} |
| CNN | Padding Pool size Activation | {same, 2} |
| LSTM | Dropout rate Timesteps Hidden nodes Learning rate Number of iteration | {relu,0.3,13,100,0.001,100} |
| LightGBM | Maximum number of leaf tree Maximum number of sample leaf node Learning rate Total number of trees constructed | {20,10,0.2,100} |

Note: total data volume = 3,492,632.

**Table 6.** Summary of prediction results by LDA-LightGBM and the comparison models.

| Model | MAE | RMSE | RSE | RAE | Coefficient of Determination | Time * |
|---|---|---|---|---|---|---|
| LSTM | 1.10 | 1.35 | 1.00 | 0.98 | 1.03% | 5765 |
| CNN | 1.10 | 1.34 | 0.99 | 0.98 | 1.25% | 1240 |
| CNN-LSTM | 1.09 | 1.34 | 0.99 | 0.98 | 2.34% | 3482 |
| LDA-LightGBM | 1.08 | 1.33 | 0.98 | 0.98 | 3.15% | 403 |

Note: * time to train the model (in seconds).

**Table 7.** Robustness evaluation (testing RMSE).

| Training Radio% | Model | Topic 1 Brand | Topic 2 Charging Speed | Topic 3 Price | Topic 4 Certification | Topic 5 Quality and Durability |
|---|---|---|---|---|---|---|
| 70 | LSTM | 1.36 | 1.35 | 1.35 | 1.35 | 1.35 |
| | CNN | 1.34 | 1.34 | 1.34 | 1.34 | 1.34 |
| | CNN-LSTM | 1.34 | 1.34 | 1.34 | 1.34 | 1.34 |
| | LDA-LightGBM | 1.34 | 1.33 | 1.33 | 1.33 | 1.33 |
| 80 | LSTM | 1.36 | 1.35 | 1.35 | 1.35 | 1.35 |
| | CNN | 1.35 | 1.34 | 1.34 | 1.34 | 1.34 |
| | CNN-LSTM | 1.35 | 1.34 | 1.34 | 1.34 | 1.34 |
| | LDA-LightGBM | 1.34 | 1.33 | 1.33 | 1.34 | 1.33 |
| 90 | LSTM | 1.36 | 1.35 | 1.35 | 1.35 | 1.35 |
| | CNN | 1.35 | 1.34 | 1.34 | 1.34 | 1.34 |
| | CNN-LSTM | 1.35 | 1.34 | 1.34 | 1.34 | 1.34 |
| | LDA-LightGBM | 1.34 | 1.33 | 1.33 | 1.34 | 1.33 |

### 3.2. Permutation Feature Importance

In this study, the predictive features of the model were trained using machine learning for permutation feature importance analysis. This analysis involves assessing the importance of customer needs and product specifications by assigning standardized score values. The permutation feature importance (PFI) method offers the following advantage: once the



prediction models are constructed, variable importance can be determined by sorting the importance scores. Generally, these scores reflect how an input variable influences decision trees within the model. Variables used more frequently in decision trees possess higher relative importance. PFI operates by iteratively shuffling individual features across the entire dataset and measuring the resulting change in the performance metric. A substantial change indicates higher feature importance.

According to Table 8, the product requirements that customers care about the most are price, charging speed, certification, quality and durability, and brand. When browsing North American e-commerce platforms, customers are particularly concerned about Wattage, size, and USB ports, based on the analysis of product specifications. These parameters should be well-known to the R&D department of the case company. Additionally, features like weight, compatible devices, and GaN are also important product features, indicating the "convenience" and "universality" of charging for the American people.

**Table 8.** Customer demands and product attributes for chargers.

| Ranking | Customer Needs |
| --- | --- |
| 1 | Price |
| 2 | Charging speed |
| 3 | Certification |
| 4 | Quality and durability |
| 5 | Brand |

| Ranking | Product attributes |
| --- | --- |
| 1 | Wattage |
| 2 | Size |
| 3 | USB ports |
| 4 | Weight |
| 5 | Compatible devices |
| 6 | GaN |

Hence, products that do not meet these characteristics will not fulfill consumer needs and will lack market competitiveness. Additionally, as shown in Tables 9 and 10, based on the correspondence between customer needs and product attributes, it is evident that compatibility and size are two product characteristics that the case company should enhance. By taking customer feedback into account, the case company can modify existing products or develop new ones, gaining a competitive edge in the market while upholding the motto of "One for All" in serving users.

**Table 9.** Key analysis of product attributes based on customer needs.

| Topic | GaN | Wattage | Compatible Devices | Size | USB-Ports | Weight |
| --- | --- | --- | --- | --- | --- | --- |
| Brand | 0.63% | 0.15% | 0.12% | 1.41% | 0.55% | 0.52% |
| Charging speed | 1.03% | 0.01% | 0.02% | 0.02% | 0.69% | 0.65% |
| Price | 0% | 0% | 0% | 0.01% | 0.01% | 0% |
| Certification | 1.11% | 0.11% | 0.11% | 0.86% | 0.75% | 0.06% |
| Quality and durability | 1.14% | 0.03% | 0.03% | 0.25% | 0.59% | 0.06% |

**Table 10.** Utility of product specifications (partial data).

| Product Attributes | Product Specification | Product Utility |
|---|---|---|
| GaN | GaN I | 133 |
| | GaN II | 66 |
| | GaN III | 25 |
| Wattage | A1 Max 20 W | 485 |
| | A2 Max 20 W | 485 |
| | A3 Max 20 W | 485 |
| | C3 Max 18 W | 284 |
| | A2 Max 12 W | 283 |
| Compatible devices | PD3.0 | 257 |
| | QC4.0 | 111 |
| | PowerIQ 3.0 | 120 |
| | PPS | 178 |
| | SCP | 69 |
| Size | 4.19 × 3.37 × 1.34 inches | 886 |
| | 3.5 × 4.5 × 1.3 inches | 501 |
| | 3.54 × 3.41 × 0.79 inches | 488 |
| | 2.09 × 1.6 × 1.17 inches | 370 |
| | 2.56 × 2.56 × 1.3 inches | 304 |
| | 2.95 × 1.42 × 1.26 inches | 264 |
| USB ports | 1C | 203 |
| | 1C1A | 57 |
| | 2C | 135 |
| | 2C1A | 250 |
| | 3C1A | 289 |
| | 1C3A | 488 |
| | 2C2A | 264 |
| | 4C | 25 |
| Weight | 2.99 ounces | 164 |
| | 4.6 ounces | 154 |
| | 13.4 ounces | 153 |
| | 2.88 ounces | 148 |
| | 2.1 ounces | 148 |

## 4. Discussion

### 4.1. Principal Findings

Manufacturers often lack immediate access to firsthand feedback from customers regarding their products, leading them to rely on "past experience" and estimations of customer preferences. As a result, there is a continued need for precise research and development (R&D) and marketing efforts. Many companies have traditionally relied on "educated guesswork" or a "trial and error" approach in product development.

To address this issue, this study integrates the LDA-LightGBM model with the quality function deployment (QFD) framework to recommend optimal product specifications. Tables 9–11 are presented within the QFD framework, as depicted in Figure 6. By translating the voice of the customer (VoC) into specific product attributes and specifications through the QFD framework, technical personnel and marketing staff can effectively communicate and integrate these analyses into their research and development processes. This ensures that the products or services introduced by a company are better aligned with customer satisfaction.

**Table 11.** 2 × 2 R&D matrix.

| Sales Volume of This Product Specification /Mainstream Market Specification | This Specification Represents the Mainstream Standard in the Target Market | This Specification Does Not Align with the Mainstream Standards in the Target Market |
|---|---|---|
| This product specification leads to higher sales volume in the target market | Continuously optimize the product specifications, while exercising caution to avoid entering oversaturated and highly competitive markets. | Conduct a cost assessment to determine its feasibility for investment in research and development. The product specification may potentially target blue ocean markets. |
| This product specification results in a lower sales volume in the target market | Conduct a cost assessment of the technical specification to evaluate its feasibility for investment in research and development. | Monitor and gather feedback from both buyers and sellers of the products. |

**Figure 6.** AI in quality function deployment (QFD).

Previous research has primarily focused on quantitative predictive models that utilize enterprise databases for product sales and customer satisfaction forecasts. However, qualitative data, such as customer feedback and specific product information from mar-

ket merchants, often go unutilized due to challenges in standardization and utilization. This means that valuable insights are overlooked in the research and development (R&D) process. To address this gap, our study employs natural language processing (NLP), specifically the latent Dirichlet allocation (LDA) model, to analyze product reviews and integrate unstructured data with structured data from Amazon's commerce platform. This approach aims to predict product ratings and identify significant variables for product outcomes across various manufacturing industries. By combining QFD and AI-based data mining, we can swiftly identify market demands and design products that truly meet consumer needs. The model assigns scores to each product feature, providing insights into their positive or negative contributions to the overall ranking of importance. Our research offers two main contributions. First, we examine the impact of integrating structured and unstructured customer feedback on product rating outcomes using a machine learning model. Second, we predict product ratings and specifications to enhance market information in product development practices. These insights improve the understanding of customer needs among R&D personnel, optimize resource allocation, and provide valuable information for decision making by customers, manufacturers, and marketing professionals. Additionally, this study assesses whether the current product specifications provided by the case company align with market trends. This forms a $2 \times 2$ product research and development matrix that can be used by manufacturing companies in Taiwan. Such knowledge is valuable for technology management theory and knowledge discovery in databases (KDD), as depicted in the table.

*4.2. Practical Implications of Sustainable Business Models*

This study examines the practical impact of data-driven sustainable business models, with a focus on two key aspects:

(1)   Data-Driven Framework for Sustainable Product Service Systems

In response to environmental regulations, companies typically adopt one of two approaches: they either react to pollution incidents with cost-effective solutions or proactively enhance materials, packaging, and product design. Unlike traditional post-development cost optimization, our study emphasizes the customization of product specifications before the development phase. While the trend of data-driven sustainable product service systems (PSS) has been extensively explored in the existing literature (Reim et al., 2015; Li et al., 2021; Pirola et al., 2020) [39–41], there is limited knowledge about how these systems operate. Furthermore, empirical research recommending sustainable product specifications and business models is lacking. Therefore, our study aims to establish a data-driven product specification recommendation system to address this critical research gap.

(2)   C2M System Framework for Sustainable Business Models

When developing a sustainable business model, companies must effectively manage material costs and introduce new models without disrupting existing product development processes. Our comprehensive approach encompasses functional value, raw material evaluation, process improvement, and the cultivation of a sustainable culture, ensuring a thorough assessment of the model's impact (Fichter et al., 2022) [42]. The C2M system framework establishes a direct connection between users and research information (Park et al., 2023) [43], facilitating precise product development for end markets, reducing waste, and minimizing consumption and redundant output. This approach can be applied to various areas related to the 2030 Sustainable Development Goals, including analyzing consumer needs, reducing material waste, promoting sustainable production practices, and developing innovative solutions for new products. It also involves exploring international certifications for sustainable product materials, including renewable materials and the development of recycling and recharging technologies. Ultimately, our research contributes to the establishment of data-driven R&D systems for building sustainable long-term business models for firms (Sharma et al., 2021) [44].

*4.3. Discuss Limitations*

This study has certain limitations. First, our experimental data are confined to the consumer electronics sector. Hence, it is important to acknowledge that the optimal predictive models proposed in our study might not universally suit diverse manufacturing domains, product categories, or experimental settings. To mitigate this, we undertook model validations with varying data proportions and compared predictions across different subjects, a strategy inspired by the approach highlighted in the academic literature. This methodology aims to enhance the robustness of our findings, especially within the constraints of industry and product category distinctions. Moreover, the generalizability and accuracy of our model warrant further validation across a wider range of industries. This necessity becomes even more pronounced when considering the diverse product types or varying scales of manufacturers. Second, the optimal model parameters we employed (as exhibited in Table 5) were meticulously adjusted to align with our specific experimental conditions. However, we acknowledge that shifts in research and development methodologies or changes in Amazon platform ratings could potentially curtail the efficacy of these parameters. Consequently, the applicability of our best-fit method to the proposed model might encounter limitations. This consideration is in line with insights from scholarly discussions on similar topics.

## 5. Conclusions

This study aims to practically apply methodologies and customer demand evaluation models in the manufacturing industry for product development with a focus on sustainable business models. The contributions of this study are summarized below.

- By integrating machine learning techniques and latent Dirichlet allocation (LDA), a two-stage product feature recommendation model is developed. This model automates the generation of comprehensive product data aligned with market preferences, considering diverse customer requirements. The LDA-lightGBM analysis successfully identifies crucial features across industries, highlighting specific product demands.
- In our study, we compare various popular deep learning architectures for sequence processing from current literature. We investigate four distinct deep learning techniques, integrating them with LDA and conducting comprehensive tests across diverse review analysis scenarios. By carefully selecting the best-performing predictive model, our proposed ensemble model demonstrates superior predictive power and robustness compared to other baseline models. This not only contributes a novel ensemble approach to the sentiment analysis modeling field but also establishes our model as a benchmark in the literature.
- Moreover, feature importance assigns a distinct score to each feature, reflecting their influence on model performance. These scores gauge the contribution of individual features to decision making within the constructed model. The analysis framework assesses product specifications across diverse consumer requirements. The integration of product specification scores and consumer needs constitutes a data-driven C2M business model, facilitating informed decision making in profitable product development.

This study aims to develop a product service system (PSS) that utilizes data analysis to understand consumer demands, extract crucial product features from customer ratings, and assess the alignment of current product specifications with market preferences. To enhance knowledge management, the system necessitates regular updates of product keywords and key specifications to prevent the extraction of irrelevant keywords during topic analysis. To address potential challenges related to data imbalances and sampling errors, future research could involve acquiring data and product specifications from various platforms, thereby enhancing dataset comprehensiveness and the accuracy of the analysis.

In the realm of systems innovation, the data-driven C2M business model presents abundant prospects for harnessing product specification scores and consumer preferences.

By integrating C2M system data via quality function deployment (QFD) and artificial intelligence (AI), the case company enhanced its R&D efficiency, adapting to the evolving post-pandemic consumer behavior landscape. The incorporation of C2M system data with advanced techniques, such as latent Dirichlet allocation and gradient-boosting decision trees, substantially amplifies industry research and development efficiency. This integration facilitates various applications, including the acceleration of product ideation, the customization of product specifications, and the improvement of communication among research and development personnel and project managers. This addition empowers traditional OEMs with customized specification capabilities and data-driven insights. Moreover, this enhancement expedites the adoption of sustainable solutions, fostering positive transformations within the manufacturing sector.

**Author Contributions:** Conceptualization, T.-C.W.; Methodology, T.-C.W.; Software, T.-C.W.; Validation, T.-C.W.; Investigation, T.-C.W.; Resources, T.-C.W.; Data curation, T.-C.W.; Writing—original draft, T.-C.W.; Visualization, T.-C.W.; Supervision, R.-S.G. and C.C. All authors have read and agreed to the published version of the manuscript.

**Funding:** This research received no external funding.

**Informed Consent Statement:** Informed consent was obtained from all subjects involved in the study.

**Data Availability Statement:** The data are not publicly available due to non-disclosure agreement (NDA).

**Conflicts of Interest:** The authors declare no conflict of interest.

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
