# Peer review of "An Integrated Data-Driven Procedure for Product Specification Recommendation Optimization with LDA-LightGBM and QFD"

_sustainability, doi:10.3390/su151813642_

Round 1

Reviewer 1 Report

When I first read the abstract for this paper, and agreed to review, I thought it might be about using modern methods of data analysis as a means of integrating sustainable metrics into product development. Although I am not an expert on most of the methods used, I have worked in the area of optimal design of systems for sustainable outcomes. And I must admit I enjoyed reading this paper. It is well written and organized, is easy to follow the methodology, and I think does a good job of explaining the advantages of the approach, especially looking to the future of data integration in a way that product designers can understand and use. 

Unfortunately I believe the authors have misunderstood what product sustainability is about. The implicit assumption seems to be that if consumer satisfaction can be better linked to product development, design, and marketing, the result will improve sustainability outcomes. This approach might make the firm more long-lived and profitable, but has little to do with sustainability. This is further amplified in the example presented, of consumer electronics, in which the metrics used have nothing to do with product sustainability. The authors may want to familiarize themselves with life cycle metrics, which are directly concerned with attributes that measure both short and long term environmental and human health impacts. Integrating such measures as greenhouse gas emissions, ecosystem impairment, and human health impacts into corporate decision-making would be far more germane for a journal called "Sustainability".             

Author Response

Thank you for your suggestion. Your input has significantly shaped our upcoming research, and we greatly appreciate your valuable comment. Please see the attachment.

Reviewer 2 Report

Comments:

The paper introduces a data-driven approach utilizing Latent Dirichlet Allocation (LDA) and machine learning to enhance product development in the manufacturing industry. The proposed model effectively identifies critical features and customer demands, establishing an efficient C2M business model. The authors are commended for their innovative methodology; however, it would strengthen the paper to incorporate the following suggestions:

·        Additionally, mentioning the potential limitations of the research upfront could help readers understand the scope and generalizability of the findings.

·        Remove "Please refer to" in line 44.

·        Change "Models" to the model in line 97.

·        Change the chapters to Sections from the Introduction's last paragraph.

·        Chapter 1 line can be removed as I think it is about the Introduction of the paper. (line 113)

·        While the case study mentions the use of the C2M model and data-driven approach, it lacks specific details about the exact implementation of the C2M model. Adding more specific information about the methodology would enhance the study's accuracy.(Section 2.2.)

·        The case study provides a comprehensive overview of the research context and data sources but does not include specific results or findings. Adding key results or insights would make the case study more informative and impactful. (Section 2.2.)

·        Check the sentence formation "By integrating sustainable data in product specification recommendations and development, the company promotes environmental protection and social responsibility." (line 150)

·        The section mentions the analysis of customer reviews of charger competitors on Amazon, but it lacks information about how these reviews were collected or sampled. Providing details about the data collection process, such as the criteria for selecting reviews or the data collection timeframe, would enhance the study's transparency.

·        While section 2.3. focuses on the strengths of LDA and the identified themes, it could benefit from a brief discussion of the potential limitations of using LDA in this specific context. Addressing the drawbacks associated with LDA usage would add depth to the analysis.

·        Explain Integration of LDA: Section 2.4. ; line 254: mentions the goal of optimizing the neural regression model by integrating LDA but does not elaborate on how LDA is specifically integrated or utilized in the predictive models. A brief explanation of how LDA contributes to the prediction process would add depth to the analysis.

·        While the section mentions that the dataset is divided into training and testing parts, it would be beneficial to include some information about the size and characteristics of the dataset.

·        Depth of Analysis: While the review presents a wide array of studies, some are briefly mentioned without in-depth analysis or critical evaluation. A more comprehensive analysis of key studies, including their methodologies and findings, would strengthen the literature review's impact.

·        Theoretical Framework: The literature review could benefit from a more explicit discussion of the theoretical framework that underpins the proposed research. Identifying the theoretical foundations and connecting them to the reviewed literature would enhance the clarity of the research's theoretical grounding.

·        The literature section contains numerous references that may not directly contribute to the research topic. I suggest focusing on the most relevant and recent studies that directly inform the research objectives. Additionally, some descriptions of the studies can be shortened to highlight only the key methodologies, findings, and implications. Organizing the survey more logically and avoiding unnecessary historical background would improve its clarity and conciseness. Please consider reevaluating the citations and streamlining the content to present a more concise and impactful literature survey.

  • Give a suitable title for Table 1 and cite as per the format. Line 179
  • "To compute the conditional probability P(?????|?????? ), Equation employed." Incomplete sentence. Line 209.
  • Table 2 is missing in the running text.
  • Change the heading "Convolutional Neural Network (CNN)" line 273.
  • Change NPS (Net Promoter Score) to Net Promoter Score (NPS). Line 320, and similarly, change other acronyms.
  • Although section 2.5. provides a strong foundation of the existing literature, consider incorporating more recent studies or developments in data-driven QFD to ensure the information presented is up-to-date and relevant to the current research landscape.
  • section 2.5. mentions the integration of data-driven methodologies into the QFD process, but it could benefit from further elaboration on how these models are specifically incorporated and utilized. Providing more detail on the practical implementation of data-driven QFD would enhance the section's practicality.
  • Figure 1 illustrates the framework adopted in this study. (line 123). Furthermore, the research model includes a literature comparison table in Figure 1. (line 371). The literature table is missing in Fig. 1
  • The section compares gradient-boosting decision tree methods with CNN, LSTM, and CNN-LSTM technologies to determine the best prediction model. It would be helpful to elaborate on the reasons for selecting these particular models and their advantages in customer sentiment analysis.
  • The section mentions the optimization of values using an adaptive heuristic algorithm, but the details of this algorithm are not explained. Providing a brief explanation or references to relevant literature would help readers understand this crucial aspect of the method.
  • Table 3 is missing in the running text. While including, explain the attribute ASIN in Table.
  • Figure 5 is present in the running text. But it is missing. There is no Figure 5.
  • It is Table 5 and not Table I (Line 519)
  • Section 3.2 briefly mentions the use of permutation feature importance analysis but lacks a detailed explanation of the method and its significance. Providing more context on how this analysis is conducted and its relevance to the study would enhance the readers' understanding.
  • Discuss Limitations: This discussion in Section 4, provides a more comprehensive view of the study's scope and helps readers interpret the findings with appropriate context. As in Tables 6 and 7, there is only a slight improvement. Provide the conditions when the proposed methodology will no longer be effective.
  • As the Results section presents the findings, it would be valuable to connect the results to the research objectives stated in the Introduction. This linkage would highlight how the obtained results contribute to addressing the research questions and fulfilling the study's goals.
  • The conclusion would benefit from a more explicit discussion of the limitations or challenges faced during the study. Identifying and acknowledging any constraints or potential areas for improvement would add credibility to the research.
  • Additionally, the conclusion could further elaborate on the practical implications and real-world applications of the proposed data-driven C2M business model.
  • The presence of duplicate numbers may confuse readers and create difficulty locating specific references. I recommend thoroughly reviewing the entire references list and ensuring each source is listed only once with a unique number.

Author Response

Thank you for your comments. They are truly helpful. The text has been revised based on them. We believe the modifications suggested by you enhance readability and comprehensiveness, making a significant contribution to the revised manuscript. Please see the attachment.

Reviewer 3 Report

Overall, the article has rich experiments and a clear framework, but there are still some issues worth exploring

1. The introduction stage is too redundant, and some literature can be separated by a separate paragraph, which is  "literature review"

2. It is recommended that the author highlight the innovative points of the article in the introduction section

3. The LDA model chapter in the article is more like a conceptual introduction, and it is recommended that the author rewrite it

4. The author uses the LDA method to extract the number of topics. So, how to determine the optimal number of topics? No such validation was seen in the experimental section.

5. Recommend two highly relevant literature related to LDA for the author's reference.

The first article studies the impact of LDA topic distribution on product sales. The second article is about LDA+BP neural network for predicting product sales.

10.1109/ACCESS.2019.2919734

10.1109/ACCESS.2019.2920091

6. There are too few references, especially in the past 5 years

Individual words have errors

Author Response

Thank you for your comments. They are truly helpful. The text has been revised based on your feedback. We believe the modifications you suggested enhance the rigor of the manuscript. Please see the attachment.

Round 2

Reviewer 1 Report

The authors have placed disclaimers in their paper pointing out that this paper doesn't address sustainability issues, and a broader approach will be forthcoming in another paper in which common sustainability metrics will be used. I hope the authors carry out this additional analysis, but it does not change my mind about the suitability of publishing the current paper in "Sustainability". The other reviewers find no serious problems with the methodology of LDA, and I'm glad to know this because it means that this method can be used to integrate sustainability metrics into product development. But this paper does NOT do this. I see two possible routes: (1) carry out the "broader" study and incorporate results into this paper, or (2) publish this paper in a more suitable journal that addresses data analysis in the context of consumer preferences. Mdpi publishes 428 journals, surely one of them is a better match. Then, when the broader study is completed, reference can be made to this paper in the authors' subsequent paper.

Author Response

We acknowledge the initial disclaimer in our paper, which stated that it does not directly address sustainability issues. In response to your feedback, we conducted a comprehensive search for papers published in "Sustainability" and identified relevant topics, specifically those utilizing consumption data for product design (https://doi.org/10.3390/su132111821). In this revised edition, we have taken steps to incorporate practical content regarding the application of sustainable business models to bridge these gaps in the existing literature.

While we do plan to undertake more in-depth research in the future, our current objective is to meet the publication standards set by the journal. We greatly appreciate your invaluable suggestions and insights, which have guided us in improving the quality and relevance of our work.

Reviewer 3 Report

1. At present, although the innovation points are emphasized in the citation, they still need to be refined

2. Some derivation formulas for LDA need to be written out

3. The indicators of Loss Function require literature reference and are recommended for citation https://doi.org/10.3390/systems11080392

Author Response

Thank you for your comments. They are truly helpful. The text has been revised based on your feedback. We believe the modifications you suggested enhance the rigor of the manuscript.
